# Virulence Factors of the Gut Microbiome Are Associated with BMI and Metabolic Blood Parameters in Children with Obesity

S. M. Murga-Garrido,[a,b] E. J. Ulloa-Pérez,[c] C. E. Díaz-Benítez,[a] Y. C. Orbe-Orihuela,[a] F. Cornejo-Granados,[d] A. Ochoa-Leyva,[d] A. Sanchez-Flores,[e] M. Cruz,[f] A. C. Castañeda-Márquez,[a] T. Plett-Torres,[b] A. I. Burguete García,[a] A. Lagunas-Martínez[a]

aCentro de Investigación en Enfermedades Infecciosas, Instituto Nacional de Salud Pública, Cuernavaca, Mexico
bPECEM (MD/PhD), Facultad de Medicina, Universidad Nacional Autónoma de México, Mexico City, Mexico
cDepartment of Biostatistics, University of Washington, Seattle, Washington, USA
dDepartamento de Microbiología Molecular, Instituto de Biotecnología, Universidad Nacional Autónoma de México, Cuernavaca, Mexico
eUnidad Universitaria de Secuenciación Masiva y Bioinformática, Instituto de Biotecnología, Universidad Nacional Autónoma de México, Cuernavaca, Mexico
fUnidad de Investigación Médica en Bioquímica, Centro Médico Nacional Siglo XXI, Instituto Mexicano del Seguro Social, Mexico City, Mexico

A. I. Burguete García and A. Lagunas-Martínez contributed equally to this work. Author order was determined by relative overall contributions.

**ABSTRACT** The development of metabolic diseases is linked to the gut microbiota. A cross-sectional study involving 45 children (6 to 12 years old) was conducted to investigate the relationship between gut microbiota and childhood obesity. Anthropometric and metabolic measurements, food-frequency questionnaires (FFQs), and feces samples were obtained. Using the body mass index (BMI) z-score, we categorized each participant as normal weight (NW), or overweight and obese (OWOB). We determined 2 dietary profiles: one with complex carbohydrates and proteins (pattern 1), and the other with saturated fat and simple carbohydrates (pattern 2). The microbial taxonomic diversity and metabolic capacity were determined using shotgun metagenomics. We found differences between both BMI groups diversity. Taxa contributing to this difference, included *Eubacterium* sp., *Faecalibacterium prausnitzii*, *Dialister*, *Monoglobus pectinilyticus*, *Bifidobacterium pseudocatenulatum*, *Intestinibacter bartlettii*, *Bacteroides intestinalis*, *Bacteroides uniformis*, and *Methanobrevibacter smithii*. Metabolic capacity differences found between NW and OWOB, included the amino acid biosynthesis pathway, the cofactor, carrier, and vitamin biosynthesis pathway, the nucleoside and nucleotide biosynthesis and degradation pathways, the carbohydrate-sugar degradation pathway, and the amine and polyamine biosynthesis pathway. We found significant associations between taxa such as *Ruminococcus*, *Mitsuokella multacida*, *Klebsiella variicola*, and *Citrobacter* spp., metabolic pathways with the anthropometric, metabolic, and dietary data. We also found the microbiome's lipooligosaccharide (LOS) category as differentially abundant between BMI groups. Metabolic variations emerge during childhood as a result of complex nutritional and microbial interactions, which should be explained in order to prevent metabolic illnesses in adolescence and maturity.

**IMPORTANCE** The alteration of gut microbiome composition has been commonly observed in diseases involving inflammation, such as obesity and metabolic impairment. Inflammatory host response in the gut can be a consequence of dietary driven dysbiosis. This response is conducive to blooms of particular bacterial species, adequate to survive in an inflammatory environment by means of genetical capability of utilizing alternative nutrients. Understanding the genomic and metabolic contribution of microbiota to inflammation, including virulence factor prevalence and functional potential, will contribute to identifying modifiable early life exposures and preventive strategies associated with obesity risk in childhood.

**KEYWORDS** childhood obesity, gut microbiota, low-grade inflammation, dietary pattern, macronutrient, virulence factors

Address correspondence to A. Lagunas-Martínez, alagunas@insp.mx.

The authors declare no conflict of interest.

Key bacteria from gut microbiota play essential roles in host dietary digestion, energy harvest, and low-grade inflammation induction, which are all involved in obesity (1, 2). Several studies have shown that gut bacterial diversity and functional capacity are different between lean and obese individuals (3–5). Host genetic factors, age, gender, immunity, and diet are among the main factors that modify the gut microbiome (6). Variations in dietary macronutrient intake impact the structure of the microbial community and its interaction with the host, affecting physiological processes (7, 8).

Excess lipid accumulation, low-grade inflammation, and other related metabolic disorders have been associated with different gut microbiome profiles (9, 10). Compositional differences in the gut microbiota between obese and lean children start at the phylum level (11). Moreover, body mass index (BMI) has been associated with several taxa, including *Eubacterium*, *Coprococcus catus*, *Ruminococcus*, *Bifidobacterium*, *Roseburia*, *Bacteroides*, and *Methanobrevibacter* (3, 12).

The gut microbiome has genes involved in dietary energy harvest, liver lipogenesis, and hormonal appetite regulation (13, 14). A collection of gut bacteria have the ability of fermenting dietary carbohydrates into short-chain fatty acids (SCFAs), which can stimulate adipogenesis (15). Other differences in microbiome functional capacity associated to obesity, includes superoxide reductase capacity, membrane transport functions, butyrate production, vitamins, and nucleotide metabolism (5, 16, 17). In addition, the gut microbiota can contribute to a low-grade systemic inflammatory state of fat deposition through several molecular interactions regulating innate immunity and inflammatory signaling (most notably *TNF-α*), indirectly participating in the onset of obesity (18).

Gut microbiome structure promotes a natural defense against exogenous pathogens and indigenous pathobionts through space competition, maintenance of the mucosa, and cross talk with the immune system that can elicit or prevent inflammation (19, 20). Colonization resistance mediating commensal-commensal and commensal-pathogen interactions is achieved through several mechanisms, including competition for nutrients or by inducing the production of inhibitory substances, such as bacteriocins (21–23). In addition, many bacteria produce siderophores and modified enterobactins (salmochelins) in order to acquire ferric iron (21, 24). Endotoxins, such as lipopolysaccharides (LPS) derived from the outer cell membrane of Gram-negative bacteria, can cross the gastrointestinal mucosa through leaky tight junctions or by infiltrating chylomicrons triggering an innate immune response when they reach systemic circulation (18, 25). The chemical structure of LPS varies across species. It is generally composed of a lipid anchor (lipid A), a core oligosaccharide region, and a polysaccharide repeating unit called the O antigen (26). The LPS variant, lipooligosaccharide (LOS), has an elaborate oligosaccharide core in place of the O antigen (26). High concentrations of systemic LPS have been called metabolic endotoxemia (27). Local inflammation is also elicited by delivery of bacterial metabolites to immune cells in the gut epithelial tissue (28). Moreover, inflammation-driven blooms of specific taxa foster horizontal gene transfer (HGT) due to higher densities of those communities in the inflamed gut, which in turn may enhance selection of strains with higher pathogenic potential and the emergence of pathobionts (28, 29). These are part of a variety of functions that influence the ability of the bacteria to grow and colonize, leading to effects in the host. Subacute or chronic inflammation, a consequence of the persistence of bacteria and immune system cross talk, may subsequently drive the development of metabolic diseases (19).

Obesity is a chronic disease with significant medical, social, and economic consequences (30). It has been reported that the combination of diet and dysbiosis results in the activation of several metabolic pathways that play essential roles in maintaining aberrant environments in the onset of obesity. Therefore, we investigated the characteristics of a subsample of overweight and obese Mexican children's gut microbiome taxonomy, functional capacity, and virulence factors loads, and its associations with anthropometric data, metabolic profiles, and dietary patterns.

**TABLE 1** General characteristics by BMI status of children from Mexico City[a]

| N = 45 | BMI status | |
| --- | --- | --- |
| Characteristics | NW (n = 26) | OWOB (n = 19) |
| Gender | | |
| Female (n = 21) | 12 (57%) | 9 (43%) |
| Males (n = 24) | 14 (58%) | 10 (42%) |
| | Avg (95% CI) | Avg (95% CI) |
| Age (yrs) | 8.04 (7.35, 8.73) | 9.05 (8.14, 9.97) |
| wt (kg)**** | 25.47 (22.66, 28.27) | 41.77 (36.17, 47.37) |
| ht (cm)* | 1.28 (1.23, 1.33) | 1.36 (1.31, 1.42) |
| Waist:Hip ratio* | 0.82 (0.81, 0.84) | 0.86 (0.84, 0.88) |
| Glucose (mg/dL) | 81.15 (76.46, 85.84) | 81.84 (77.94, 85.75) |
| Triglycerides (mg/dL)● | 75.23 (65.16, 85.31) | 105.68 (84.61, 126.76) |
| Cholesterol | | |
| HDL (mg/dL) | 50.85 (46.98, 54.71) | 49.89 (44.54, 55.25) |
| LDL (mg/dL) | 100.88 (89.86, 111.91) | 109.89 (95.02, 124.77) |
| Total (mg/dL) | 152.96 (138.37, 167.55) | 168.05 (148.58, 187.53) |
| Blood pressure | | |
| Systole (mean) | 93.85 (89.9, 97.79) | 99.84 (95.49, 104.2) |
| Diastole (mean) | 64.38 (60.86, 67.9) | 65.39 (61.97, 68.81) |
| Physical activity (Mets) | 348.2 (233.62, 462.78) | 387.79 (182.69, 592.9) |
| Total Energy (Kcal) | 3032.89 (2661.05, 3404.74) | 2768.62 (2409.88, 3127.36) |
| Family history of overweight/ obesity (%) by BMI group | 50% | 68% |

[a]Mean comparison test. ns: $P > 0.05$; ●, $0.1 > P > 0.05$; *, $P \leq 0.05$; **, $P \leq 0.01$; ***, $P \leq 0.001$; ****, $P \leq 0.0001$. $P$ values adjusted for multiple comparisons (FDR).

## RESULTS

**Overview of the urban children population and classification of the metabolic parameters.** This study included a subsample of 45 individuals 6 to 12 years old from a cross-sectional study previously published (31). Individual information collected from healthy participants, included dietary frequency questionnaire and phenotypic characterization including age, gender, and BMI. Participants were grouped into normal weight (NW) or overweight and obese (OWOB), using the World Health Organization (WHO) z-scores for childhood BMI adjusted by gender and age (32). Our study included 21 female participants, from which 12 (57%) were classified as normal weight and 9 (43%) as overweight and obese. Meanwhile, of the 24 male participants, 14 (58%) were classified as normal weight and 10 (42%) as overweight and obese (Table 1).

The mean age was 8 years old for the normal weight group and 9 years old for the group with overweight and obesity. Weight (kg), height (cm), and waist-to-hip ratio (W:H) resulted statistically different between both BMI classifications. The mean for NW participants was 25.5 kg, 128 cm, 0.82, and 75.2 mg/dL for weight, height, W:H, and triglycerides respectively; whereas, for OWOB children, these values increased to 41.8 kg, 136 cm, 0.86 W:H, and 106 mg/dL. Blood glucose, triglycerides, cholesterol, and blood pressure (BP) values were not statistically different, although the OWOB group showed higher mean values compared to the NW group.

**Two dietary profiles were predominant throughout the urban children population.** As a result of the analyses of each participant's dietary data collected from a food frequency questionnaire (FFQ) (Tables S1, S2, and Fig. S1), 2 dietary patterns were identified and interpreted as "Pattern 1," characterized by the consumption of proteins and complex carbohydrates, and "Pattern 2," characterized by the consumption of saturated fat and simple carbohydrates (Table S3). Based on the reported diet from each child, we derived a score to approximate which pattern their diet was more likely to be characterized by (Table S4).

**Taxonomical microbiome diversity in NW and OWOB groups.** We obtained a total of 1,617,251,896 high quality microbiota reads from fecal samples. Taxonomic and

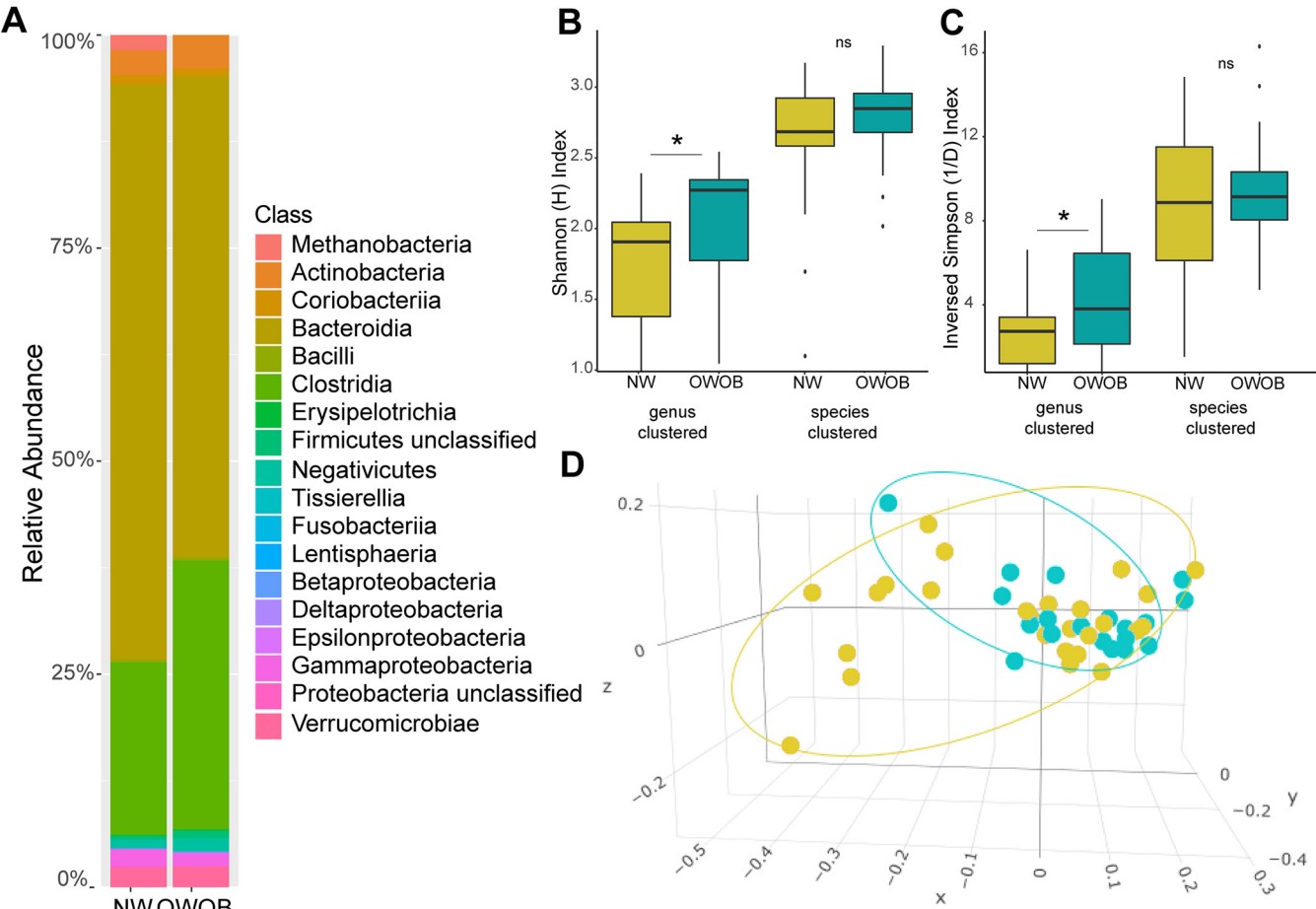

**FIG 1** Microbiota diversity. (A) Class relative abundance of the 45 samples included in this study grouped by BMI. (B and C) Alpha diversity clustered by genera and species. (B) Shannon Index 'H' and (C) Inversed Simpson diversity Index '1/D' of normal weight and obese-overweight groups. NW group is shown in gold color, and OWOB in teal. ns: $P > 0.05$; *, $P < 0.05$. Pairwise comparisons were obtained using Wilcoxon Ranksum Test. (D) Non-metric multidimensional scaling (2-axis stress plot value = 0.089) of weighted UniFrac distance between NW group (gold) and OWOB (teal).

functional annotation was performed using clade-specific marker genes (Fig. 1A). We annotated 291 species belonging to 120 genera, 52 families, and 29 orders throughout the 45 samples. We found 247 species with a relative abundance over 0.01%. The most abundant genera, include *Bacteroides*, *Prevotella*, *Alistipes*, *Ruminococcus*, *Eubacterium*, *Faecalibacterium*, *Bifidobacterium*, *Roseburia*, *Akkermansia*, and *Barnesiella*.

We analyzed the population's diversity by comparing NW and OWOB based on the fact that microbiota dysbiosis favors an inflammatory status and impairs energy metabolism, leading to an obese phenotype (33). As a result, Shannon and inverse Simpson alpha diversity indexes, clustered by genera, between NW and OWOB groups, were significantly different (Wilcoxon Rank-sum Test (WRST), $P$ value $< 0.05$) (Fig. 1B and C). However, the alpha diversity difference between groups was not significantly clustered by species.

Non-metric multidimensional scaling (NMDS) of weighted Unifrac (wUF) distance, a metric sensitive to taxonomic phylogenetic distances that considers species abundance, showed a clustering by BMI (adonis-PERMANOVA $P$ value $< 0.05$) (Fig. 1D).

Taxa that most contribute to beta variation (more than 70%) between NW and OWOB groups (Table S5), obtained by a similarity of percentages analysis (SIMPER), include genera *Eubacterium*, *Faecalibacterium* (WRST $P$ value = 0.08 and 0.04, respectively) and species *Bacteroides uniformis* ($P$ value = 0.04), *Methanobrevibacter smithii* ($P$ value = 0.08), *Eubacterium* sp ($P$ value = 0.02), and *Faecalibacterium prausnitzii* ($P$ value = 0.04) (Fig. 2A).

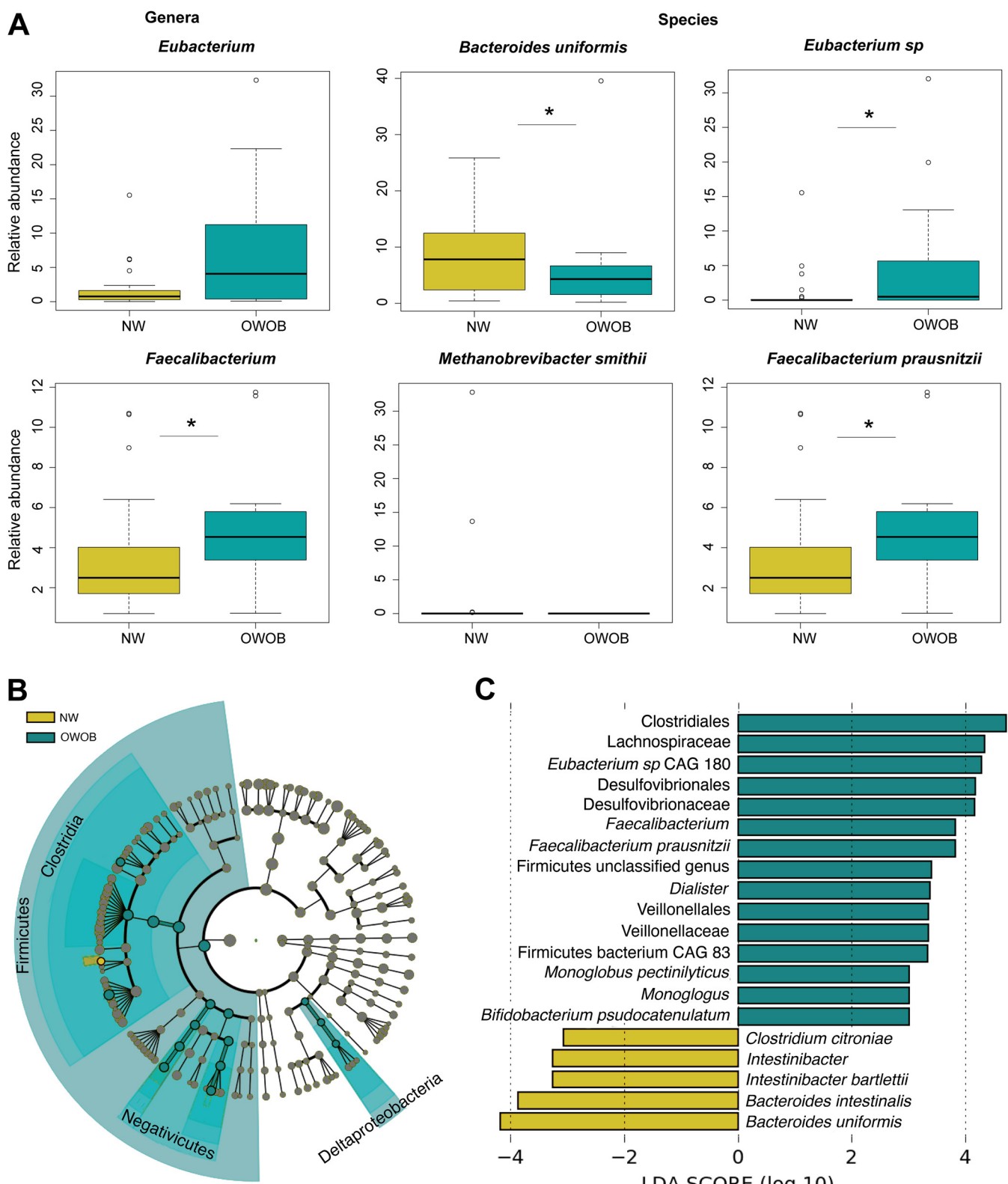

**FIG 2** Taxa difference between BMI groups. (A) Relative abundance of taxa that showed differences in the normal weight (NW, gold) group compared to overweight-obese (OWOB, teal). Pairwise comparisons using Wilcoxon Rank-Sum Test adjusted by FDR: *Eubacterium* P value = 0.08, *Faecalibacterium* P value = 0.04, *Bacteroides uniformis* P value = 0.04, *Methanobrevibacter smithii* P value = 0.08, *Eubacterium* sp. P value = 0.02, *Faecalibacterium prausnitzii* P value = 0.04. *, P ≤ 0.05. (B and C) Bacterial taxa contributing to differences between NW and OWOB microbiomes obtained using Linear discriminant analysis Effect Size (LEfSe) analysis. (B) Cladogram showing the comparison result for the NW (gold) and OWOB (teal) groups, where colors distinguish taxa differences between both. (C) List of taxa differentially abundant between NW and OWOB microbiome communities (P value < 0.05), where the LDA score (log 10) is indicated at the bottom.

We used a linear discriminant analysis effect size (LEfSe) algorithm to identify bacterial features that most likely explain differences between NW and OWOB classes, taking into account biological consistency and effect relevance (34). OWOB group showed a higher abundance of the phylum Firmicutes and classes, such as Clostridia, Negativicutes, and *Deltaproteobacteria* (Fig. 2B). Taxa that most likely explain the differences between the OWOB group and the NW, include *Eubacterium* sp., *Faecalibacterium prausnitzii*, *Dialister*, *Monoglobus pectinilyticus*, and *Bifidobacterium pseudocatenulatum*. *Clostridium citroniae*, *Intestinibacter bartlettii*, *Bacteroides intestinalis*, and *Bacteroides uniformis* were found to have a significant effect size in the NW group (LDA scores [$\log_{10}$] < −2 and >2, *P* value < 0.05) (Fig. 2C). Other taxa marginally differential between both groups are shown in Fig. S2 (LDA scores [$\log_{10}$] < −2 and >2, *P* value between 0.05 and 0.1).

**Microbiomes' metabolic potential differences.** We obtained 436 annotated pathways as resumed by MetaCyc (Table S6). Differential pathway abundance yielded higher LDA scores (LDA scores [$\log_{10}$] > 2, *P* value < 0.05) in the OWOB group compared to NW, in the amino acid biosynthesis pathway (L-arginine biosynthesis, L-ornithine biosynthesis); the cofactor, carrier, and vitamin biosynthesis pathway (coenzyme A and pantothenate biosynthesis, thiamine formation pathways); the nucleoside and nucleotide biosynthesis and degradation pathways (UMP biosynthesis, 5-aminoimidazole ribonucleotide biosynthesis, pyrimidine deoxyribonucleotides *de novo* biosynthesis, purine ribonucleoside degradation); the carbohydrate-sugar degradation pathway (galactose degradation, stachyose degradation); and the amine and polyamine biosynthesis pathway (norspermidine biosynthesis) (Fig. 3). Pathways with significantly different abundance in NW group, included the amino acid degradation and fermentation pathway (l-glutamate degradation VIII, succinate fermentation to butanoate); the cofactor, carrier, and vitamin biosynthesis pathway (superpathway of demethylmenaquinol biosynthesis, superpathway of menaquinol biosynthesis, gamma-glutamyl cycle); the amine and polyamine biosynthesis pathway (superpathway of arginine and polyamine biosynthesis); and lipid biosynthesis pathway (Fig. 3).

The aforementioned differential pathways were annotated in 35 taxa genomes. The pathway abundances were contributed mainly by genera *Escherichia*, *Bacteroides*, *Faecalibacterium*, *Eubacterium*, *Bifidobacterium*, *Blautia*, *Ruminococcus*, and *Roseburia* (Fig. S3).

**Anthropometric, metabolic, and dietary data are associated with specific gut microbial taxa and functional capacity.** We assessed each feature's abundances using generalized linear models (compound Poisson linear model, CPLM) to identify individual microbial taxa associated with our population's anthropometric, metabolic, and dietary covariates. We analyzed these associations using transformed relative abundance data with corrections for sparse (zero-inflated) compositional microbial feature data. Additionally, among the population's variables, we identified both age and sex to be associated with the taxonomic microbiome composition (adonis-PERMANOVA, *P* value < 0.05), in agreement with previous findings (5, 35). Therefore, we adjusted for age and sex in the analyses comprising species-level taxonomic profiles.

We found a total of 31 taxa significantly associated with W:H ratio, 28 taxa to glucose, 33 to triglycerides, and 36 to total cholesterol (*P* value, *q* value, and FDR < 0.05) (Table S7). The abundance of species included in the genera *Victivallis*, *Prevotella*, *Mitsuokella*, *Clostridium*, and the family Ruminococcaceae were overall associated with decreased values of the population's parameters. In contrast, members of *Parvimonas*, *Klebsiella*, *Gemella*, *Enterococcus*, and *Citrobacter* were positively associated with W:H ratio, blood glucose, triglycerides, and total cholesterol (Fig. 4).

Subsequently, we sought to evaluate the microbiota taxa abundance and the dietary data estimated from the FFQ. We found 46 taxa significantly associated with total energy, 40 with pattern 1 and 52 with pattern 2, 38 with proteins, 35 with fiber, 30 with carbohydrate, 39 with sugar, 31 with lipids, 25 with saturated fatty acids, 36 with trans-fatty acids, 36 with polyunsaturated fatty acids, and 37 with monounsaturated fatty acids percentage (*P* value, *q* value, and FDR < 0.05). More details can be found in Table S7. We found species of the genera *Prevotella*, *Megasphaera*, *Klebsiella*, *Gemella*, *Citrobacter*, *Bacteroides*, and

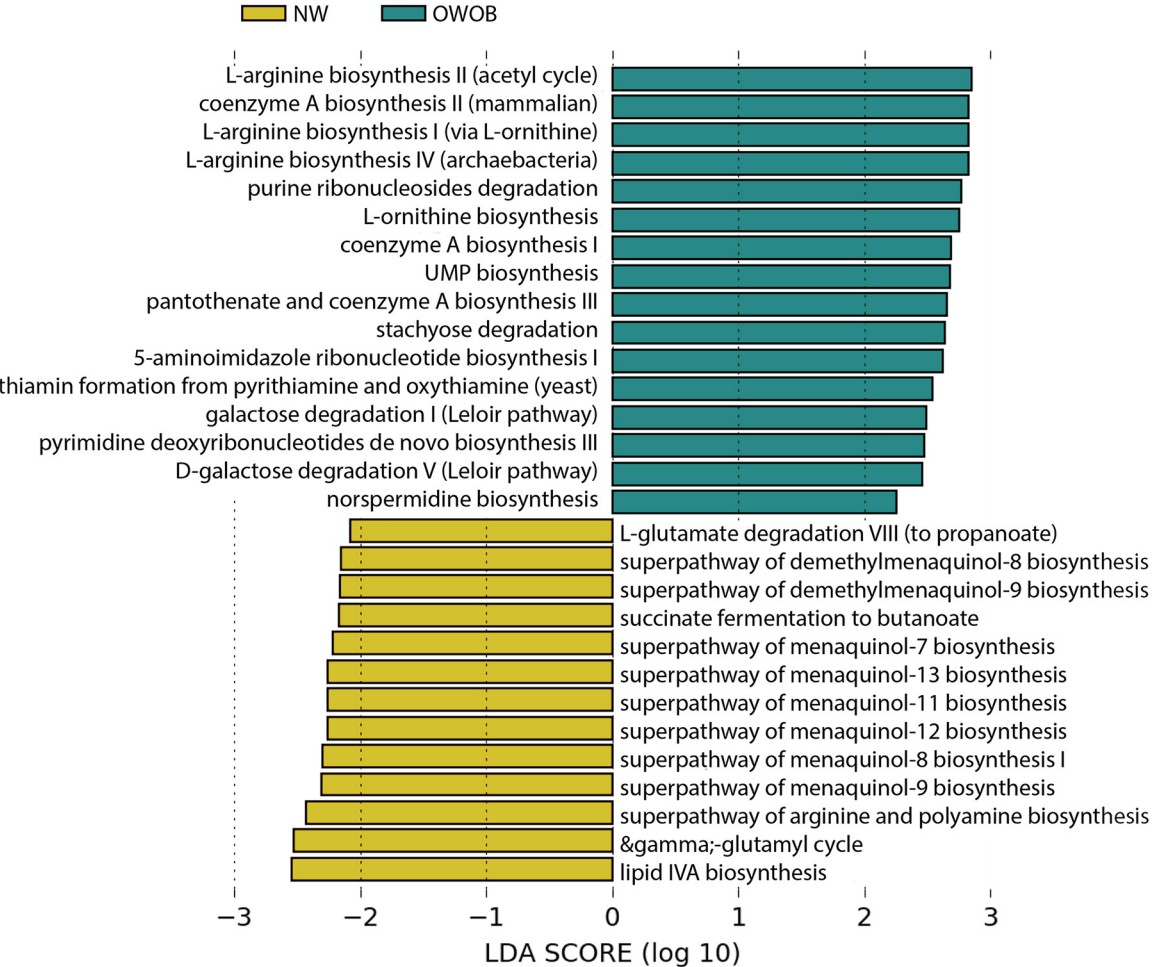

**FIG 3** Linear discriminant analysis Effect Size (LEfSe) summary. List of path- ways as summarized by MetaCyc differentially abundant (*P* value < 0.05) between microbiome community of the NW (gold) group and the OWOB (teal). LDA score (log 10) is indicated at the bottom of the graph.

*Alloprevotella* positively associated to total energy intake, sugar, lipid, saturated fatty acids, and trans-fatty acids percentage intake (Fig. 4). Interestingly, the abundance of most of these taxa were also associated with increments in the W:H ratio, the metabolic parameters, and lower protein, fiber, and unsaturated fatty acids intake percentage. Conversely, we found members of *Victivallis*, Ruminococcaceae, *Mitsuokella*, and *Clostridium* associated with higher protein, fiber, carbohydrate, monounsaturated fatty acids, and polyunsaturated fatty acids intake (Fig. 4).

Relevant taxa associated with protein and complex carbohydrates (pattern 1), or saturated fat and simple carbohydrates (pattern 2) intake included species of the taxonomical orders Lactobacillales, Clostridiales, Actinomycetales, Bacteroidales, Bifidobacteriales, Eubacteriales, Methanobacteriales, Desulfovibrionales, Actinomycetales, and Pseudomonadales (Fig. 4). It is worth noting that most taxa that were positively associated with pattern 2 showed a negative association with pattern 1.

To understand how microbial functional capacities further relate to our population's metabolic and dietary parameters, we tested MetaCyc pathways and gene families, as summarized by level 4 enzyme commission (EC) categories obtained through HUMAnN3 (36, 37). For analyses comprising functional capacity summarized as MetaCyc, we adjusted by physical activity (adonis-PERMANOVA, *P* value < 0.05), and for ECs, we adjusted by age (adonis-PERMANOVA, *P* value < 0.1) (Table S8). The strongest associations with BMI, blood triglycerides, total cholesterol, trans-fatty acids percentage, and dietary pattern 2 included 3 pathway clusters and 6 gene family clusters (Table 2 and Table 3). The dietary trans-fatty

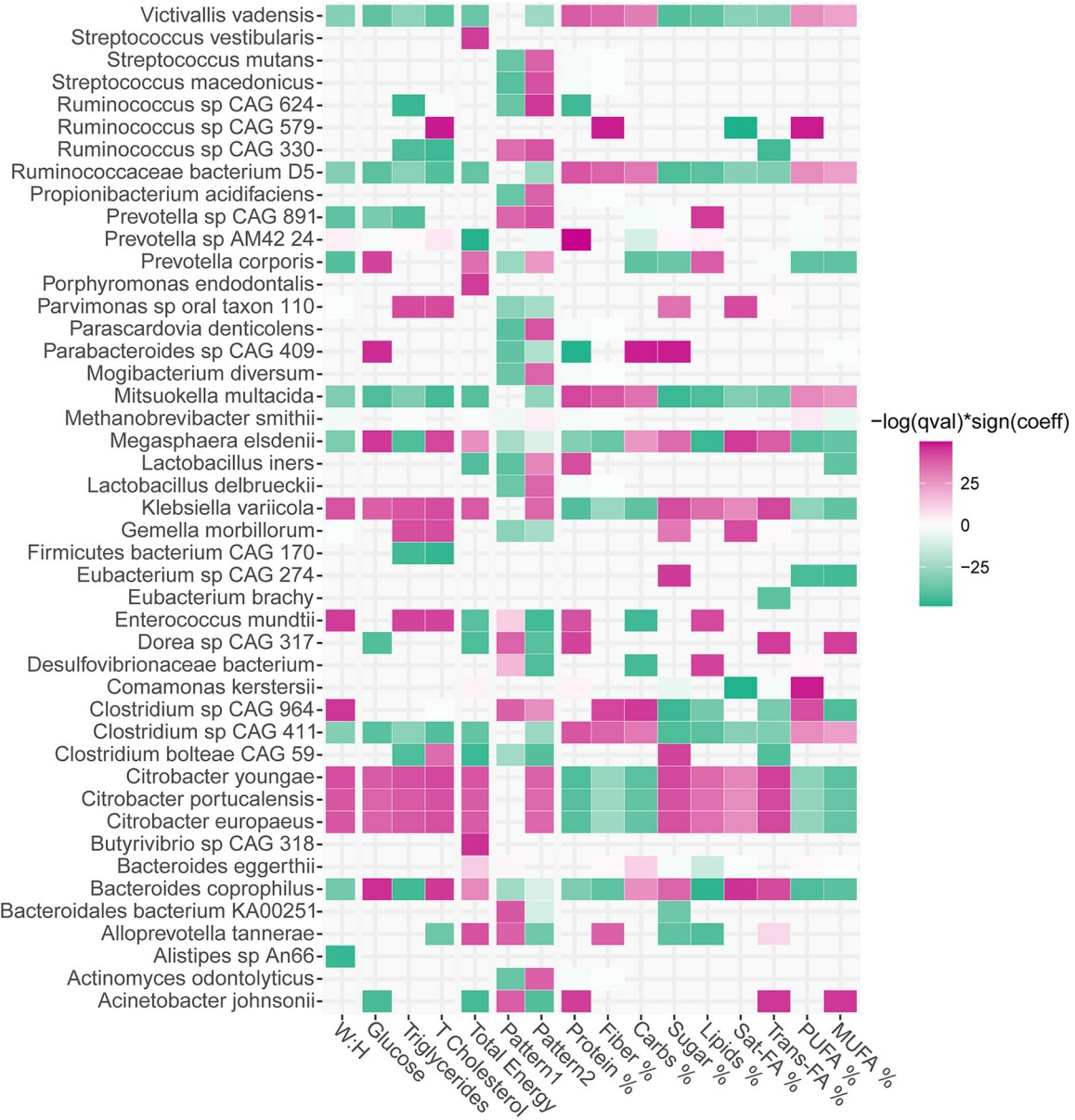

**FIG 4** Heatmap indicating significant associations (*P* value < 0.05) found between our population microbiome taxa abundance with the anthropometric data of waist-to-hip ratio (waist:hip); the metabolic blood parameters of glucose (mg/dL), triglycerides (mg/dL), and total cholesterol (mg/dL); and the dietary data obtained from FFQs including intake percentage of proteins, fiber, carbohydrates, sugar, lipids, saturated fatty acids, (S-FA), trans-fatty acids, (T-FA), polyunsaturated-fatty acids (PUFA), and monounsaturated fatty acids (MUFA). Associations were obtained using Multivariate Association with Linear Models (MaAsLin2). List of taxa, coefficients, standard error, *P* values, and *q* values/FDR are listed in Supplemental Table S7.

acid percentage was positively associated with 1-4-dihydroxy-6-naphthoate biosynthesis II, and negatively associated to the gene family arabinose-5-phosphate isomerase (*P* value < 0.05, *q*-value < 0.05).

These functional associations suggest that microbiome metabolism plays a role in blood triglycerides, weight gain, and total cholesterol levels. In addition, it is related to trans-fatty acids and pattern 2 predominant dietary intake.

**Microbiomes' virulence factors diversity differs between NW and OWOB groups.** Obesity has been linked to inflammatory processes including endotoxemia related to lipopolysaccharides (38). We used ABRicate to screen assemblies for virulence factor genes and obtained 6,711 annotations throughout 426 virulence factors. After filtering by coverage (≥50%) and identity (≥75%), we were able to identify 237 virulence factors present

**TABLE 2** Multivariate Associations with Linear Models (MaAsLin2) between microbiota functional capacity summarized as MetaCyc, and population metabolic, dietary data, adjusted by physical activity[a]

| Pathway (MetaCyc) | | Effect | Coef | 95% CI | Pval | Qval |
|---|---|---|---|---|---|---|
| PWY.4041 | Gamma-glutamyl cycle | BMI z-score | −6.725e-4 | −1.01e-03, −0.33e-03 | 3.58e-4 | 0.0860 |
| PWY.6527 | Stachyose degradation | TG mg/dL | 6.4815e-4 | 2.98e-04, 9.98e-04 | 7.64e-4 | 0.0909 |
| PWY.7371 | 1-4-dihydroxy-6-naphthoate biosynthesis II* | Trans-fatty acids % | 1.006e-4 | 0.57e-04, 1.44e-04 | 5.21e-5 | 0.0230 |

[a]*, P value and q-value < 0.05.

throughout our samples (Table S9). Observed richness and Shannon alpha diversity were significantly higher in the NW group compared to the OWOB (Fig. 5A and B).

Virulence factor genes were clustered by virulence factor identifiers (VFID) for further analysis. Lipooligosaccharide cluster (LOS, CVF 494, and VF 0044) was identified as differentially abundant in the OWOB group compared to NW (log Fold Change = 8.29, P value < 0.001, FDR < 0.05) (Fig. S4, Table S10).

We evaluated how the children's metabolic parameters and dietary data related with the virulence factors categories found in their microbiome (Table S11). We observed 5 categories of virulence factors, comprising exotoxins, effector delivery systems, adherence, immune modulators, and nutritional or metabolic factors associated with the BMI, W:H ratio, blood glucose, triglycerides, total cholesterol, and different dietary intake (Fig. 5C). The iota-toxin and heat-stable toxin (ST), both part of the exotoxin category, were associated with the BMI and W:H ratio. The metabolic parameters of blood total cholesterol, glucose, and triglycerides resulted in the highest amounts of significant associations with the microbiome's virulence load. Seven virulence factors clustered by VFID comprised in all 5 virulence factors categories were positively associated with pattern 1 consumption. Six VFID showed a negative association with sugar percentage intake. Lipid, trans-fatty acids, and monounsaturated fatty acids intake percentages showed associations with the different factors of the exotoxin category.

## DISCUSSION

Overweight, obesity, and associated metabolic syndrome in childhood have been increasing worldwide (39). Mexico is among the countries with higher childhood obesity rates (40). In our study, participants grouped as OWOB showed significantly higher visceral abdominal fat accumulation, weight, and marginally higher triglycerides, suggesting this group is at higher risk of developing metabolic impairment.

Differences within gut bacterial diversity, taxonomical composition, and phyla ratios between lean and obese individuals have been largely reported (2). Consistently, in this study, we found significant differences between the alpha and beta diversity in the gut microbiota of NW and OWOB, supporting the hypothesis that certain phyla enhance higher energy harvest (4).

The abundance of taxa found in this study, such as Porphyromonadaceae, *Alistipes*, *Bacteroides*, *Clostridium citroniae*, *Intestinibacter bartlettii*, and *Methanobrevibacter smithii*, have been previously reported higher in non-obese individuals (41, 42). We also report species associated with healthier metabolic parameters and dietary patterns including *Victivallis vadensis*, *Ruminococcus* sp., *Mitsuokella multacida*, *Clostridium* sp., *Alistipes* sp., and *Acinetobacter johnsonii*. Previous studies have associated these bacteria with anti-

**TABLE 3** Multivariate Associations with Linear Models (MaAsLin2) between microbiota gene families, as summarized by level 4 ECs, and population metabolic and dietary data adjusted by age[a]

| Gene family EC-level4 | | Effect | Coef | 95% CI | Pval | Qval |
|---|---|---|---|---|---|---|
| X4.3.1.1 | Aspartate ammonia-lyase | BMI z-score | −0.382 | −5.39e-01, −2.26e-01 | 2.19e-05 | 0.073 |
| X2.7.7.77 | Molybdenum cofactor guanylyltransferase | Cholesterol Total mg/dL | 0.318 | 1.56e-01, 4.80e-01 | 0.04e-02 | 0.090 |
| X3.4.13.21 | Dipeptidase E | Cholesterol Total mg/dL | 0.287 | 1.40e-01, 4.35e-01 | 0.04e-02 | 0.090 |
| X1.1.1.1 | Alcohol dehydrogenase | Cholesterol Total mg/dL | 0.259 | 1.25e-01, 3.94e-01 | 0.05e-02 | 0.093 |
| X1.2.7.1 | Pyruvate synthase | Pattern 2 | 0.415 | 2.46e-01, 5.84e-01 | 1.96e-05 | 0.066 |
| X5.3.1.13 | Arabinose-5-phosphate isomerase* | Trans-fatty Acids % | −0.698 | −9.63e-01, −4.33e-01 | 6.33e-06 | 0.014 |

[a]*, P value and q value < 0.05.

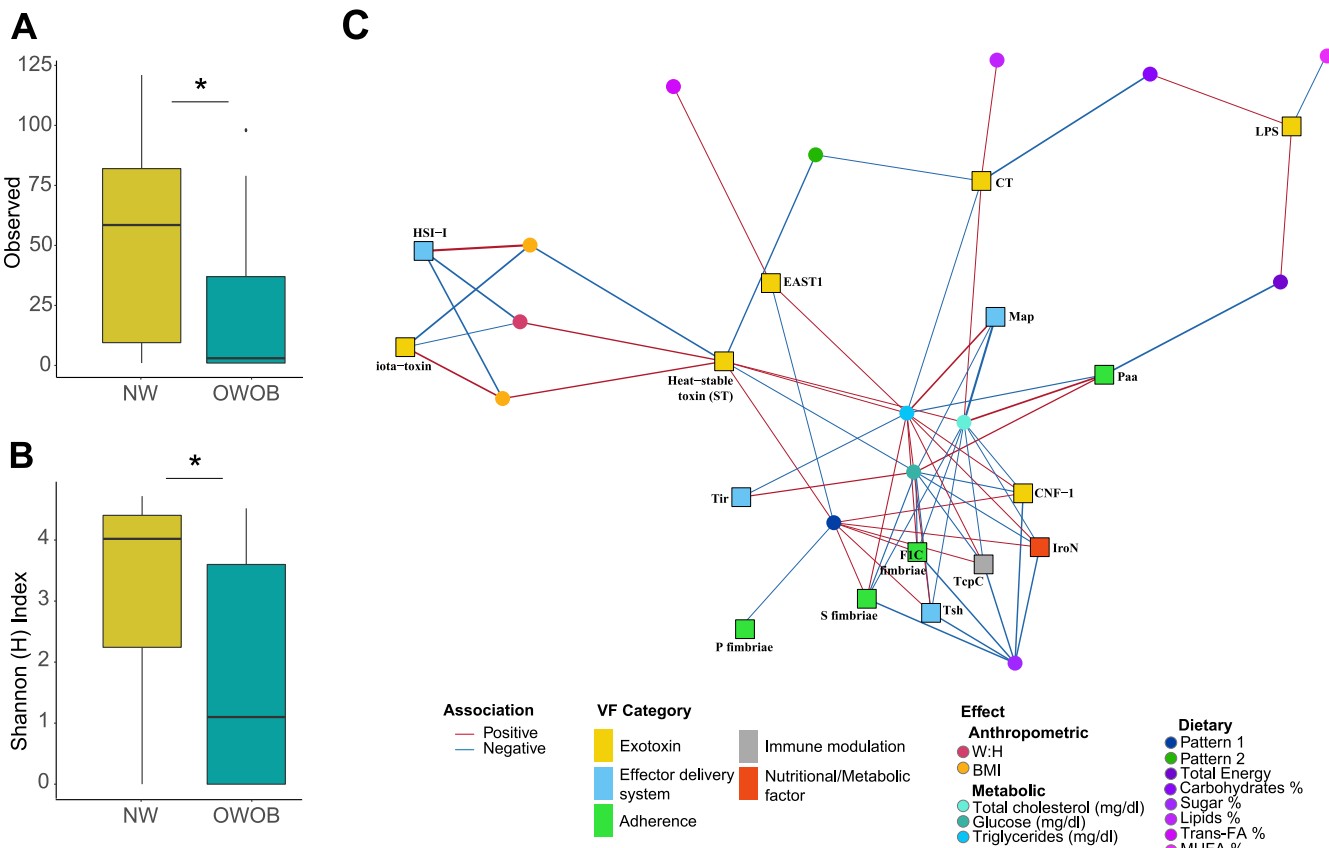

**FIG 5** Virulence factors, (A) richness and (B) Shannon Index 'H', showing alpha diversity between NW (gold) and OWOB (teal) groups. *, P value < 0.05. Pairwise comparisons were obtained using WRST. (C) Association network between virulence factors clustered by VFID and anthropometric data, metabolic blood parameters, and dietary intake obtained from FFQs. Virulence factors categories are indicated in yellow (exotoxins), blue (effector delivery system), green (adherence), gray (immune modulation), and red (nutritional or metabolic factors). Associations were obtained using Multivariate Association with Linear Models (MaAsLin2). List of VFID, coefficients, standard error, P values, and q values are listed in Table S11. BMI, Body Mass Index; W:H: waist-to-hip ratio; Glu: Glucose; TG: Triglycerides; TC: Total Cholesterol; T-FA:;trans-fatty acids; MUFA, monounsaturated-fatty acids.

inflammatory and pain relieving processes, and with non-digestible carbohydrates and protein intake (43–46). Likewise, the abundance of Veillonellaceae, *Eubacterium*, *Dialister*, *Lactococcus*, *Roseburia*, *Fusicatenibacter saccharivorans*, *Coprococcus catus*, *Bilophila*, *F. prausnitzii*, and *Bifidobacterium*, has been reported higher in individuals with obesity and with proinflammatory statuses (41, 47, 48). In addition, in this study, higher anthropometric and blood biochemical values along with sugar and lipid (Pattern 2) intake correlated with the abundance of *Klebsiella variicola*, *Gemella morbillorum*, *Enterococcus mundtii*, *Citrobacter youngae*, *C. portucalensis*, and *C. europaeus*. These taxa have been previously reported as potential markers associated with obesity, overweight, increased fasting glucose, and inflammatory molecules (49–51). Moreover, simple carbohydrates and high-fat diets have been associated with *Klebsiella* and *Citrobacter* spp. abundances, respectively (52, 53). It is worth noting that dietary macronutrients, including fat, protein, and carbohydrates, have been associated with significant shifts in the gut microbiota (54).

Over-representation of nucleotide, amino acid, pantothenate and coenzyme A (CoA) pathways found in the functional capacity analysis in the microbiome of the OWOB group have been previously identified as more abundant in the microbiome of children, elderly, and woman with obesity, tissue deposition, and metabolic syndrome when compared to lean, healthy individuals (55–58). Bacterial functional pathways related to nucleotide and lipid metabolism are modulated by physical activity (58), which was identified as a fixed effect through our microbiome functional analysis. The carbohydrate and sugar degradation pathway in the microbiome plays a role in energy harvest, thus providing a possible explanation of the galactose and stachyose degradation abundance in the OWOB group,

and the positive association with triglycerides (59). Conversely, the most abundant end products of the gut microbiota's amino acid degradation and fermentation are SCFAs and BCFAs, that have been implicated in glucose and lipid metabolism modulation in the group of children with normal weight (60, 61). In this study, we also found Gamma-glutamyl and menaquinones metabolism negatively associated with BMI, in agreement with previous negative associations found with adiposity, liver triglycerides, glucose, and TNF-$\alpha$ signaling in mice models (62, 63). In contrast to our results, a correlation between menaquinone and BMI in obese adults from Denmark was reported (64), suggesting that factors such as age and diet may generate this discrepancy.

Interestingly, we found arabinose-5-phosphate isomerase, a bacterial gene that plays a role in synthesizing LPS for the outer membrane in Gram-negatives and a carbohydrate transport and metabolism regulator in Gram-positives, negatively associated with trans-fatty acids percentage intake (65, 66).

By recognizing signals and nutrients, gut pathobionts can coordinate the expression of their virulence traits and adjust their metabolism for an advantage in nutrient competition and colonization (67).

Circulating endotoxins have been associated with increments of cytokines such as TNF-$\alpha$ and IL-6 in adipocytes and, accordingly, subclinal levels of LPS, have been linked with high-fat diets, obesity, and other metabolic disorders (27, 68). In addition, lipooligosaccharides (LOS) can also induce proinflammatory cytokines through TLR4 (69). In this study, we found that the NW microbiome preserved a more even virulence factor alpha diversity; whereas, the OWOB group had more dominance. The LOS (CVF494, VF0044) category was overrepresented in the OWOB group compared to NW, supporting the connection between endotoxins and low-grade inflammation in the obesity process. We also identified the iota-toxin (VF0381), which binds to a target cell and ADP-ribosylates its actin, resulting in cell rounding and death (70). Heat-stable enterotoxins (ST [VF0211], EAST1 [VF0216], and cholera toxin [CT] [VF0128]) are peptides relevant to intestinal inflammation, electrolyte imbalance, diarrhea induction, and disruption of the gut microbiota (71, 72). Cytotoxic necrotizing factor 1 (CNF-1 (VF0240)) is a toxin that modulates the family of Rho GTPases into their activated GTP-bound state, leading to NF-$\kappa$B activation and release of proinflammatory cytokines and upregulation of cyclooxygenase-2 (COX-2) (73, 74) (Fig. 6). These findings suggest that bacterial toxins play a direct or indirect role in BMI increments and host metabolic impairment through inflammation, and that they are associated with pattern 1, lipids, and trans-fatty acids intake.

The effector proteins that subvert the host cell processes are delivered through translocases conforming a complex that consists of a basal-body-like structure, a neck domain, and an external needle (75). The type III secretion (T3SS) is encoded in the locus of enterocyte effacement (LEE) inside a pathogenicity island (PAI), and it is essential in the formation of attaching and effacing (A/E) lesions on enterocytes (76). Several bacterial molecules are exported by the LEE-encoded type III secretion system, such as the intimin receptor (Tir) that is secreted into the host cell, where they acts as receptors to bind intimin on the bacterial surface. Interestingly, changes in the glucogenic conditions or the production of fucose or succinate from host mucin by gut microbiota modulates the expression of the LEE genes in enteropathogens (67, 77). Gram-negative bacteria can secret serine protease autotransporters of Enterobacteriaceae (SPATEs) through the type V secretion system (T5SS), such as Tsh (temperature-sensitive hemagglutinin), that has adhesive and proteolytic functions (78, 79). Map (VF0195) exhibits a Guanine nucleotide exchange factors (GEFs) activity that regulates the Rho family of small GTPases that, in turn, induces transient filopodium formation at bacterial attachment sites (80, 81). On the other hand, HIS-I (VF0334) is part of the effector proteins translocated into an adjacent prokaryotic or eukaryotic cell by the transmembrane multiprotein employed by Gram-negative bacteria, the type VI secretion system (T6SS) (82) (Fig. 6). We found different virulence factors in this category to be mostly

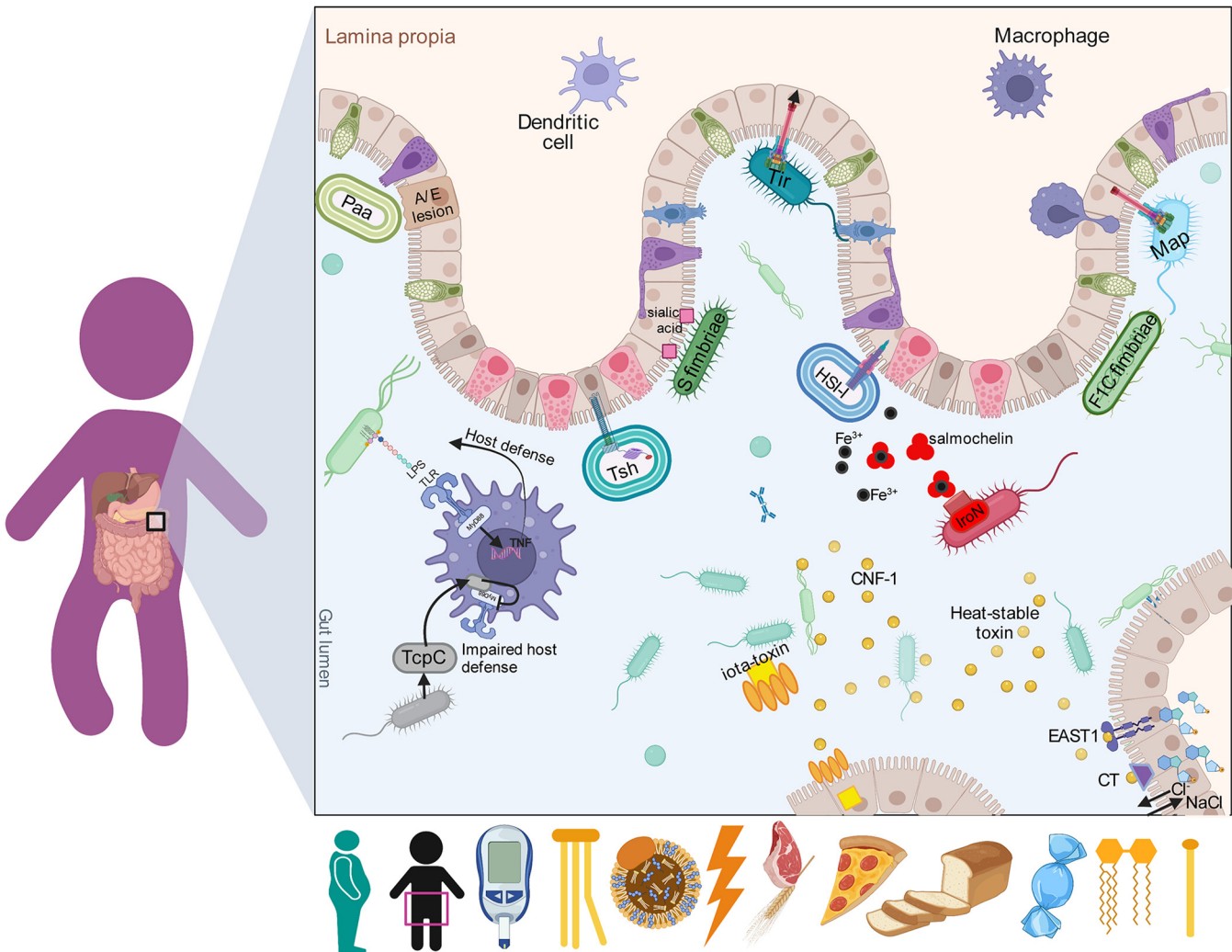

**FIG 6** Graphical summary of the gut microbiome virulence factors associated with BMI, W:H ratio, glucose, triglycerides, total cholesterol, total energy, pattern 1 and 2, sugar, lipid, and trans-fatty acids percentage intake. In this figure, we show the lamina propria with immune cells (in purple), enterocytes, Goblet cells (green), Paneth cells (pink), and the intestinal lumen. We identified virulence factors included in the categories of adhesins binding to sialic acid receptors (F1C Fimbriae, SFimbriae, and Paa, in green); bacterial molecules secreted by systems (Tir, Tsh, Map, and HIS-I, in turquoise); immunomodulatory proteins inhibiting macrophage signaling via MyD88 (TcpC, in gray); toxins promoting the liberation of ions (iota, CNF-1, thermostable, EAST1, and CT, in yellow); endotoxins (LPS, in light green); and metabolic and nutritional factors (IroN, in red).

associated with the blood metabolic parameters. To date, the biological implications of these associations are unknown.

We also found significant associations with virulence factors included in the category of adherence. Adhesins, that we found, bind to sialic acid-containing receptors (F1C fimbriae [VF0224], S fimbriae [VF0222]), and porcine A/E lesion-associated adhesin (Paa [VF0194]) associated with biochemical metabolic markers (Fig. 6); the information is still limited in relation with obesity development (83).

Under inflammatory conditions, the concentration of NO3- increases and iron concentration decreases, enhancing pathogenic and pathobiont fitness in the modified environment (84). We found IroN (VF0230), a pathogen prevalent salmochelin siderophore receptor, strongly associated with sugar intake and blood triglyceride levels. We also found TG levels associated with TcpC (VF0413). The immunomodulatory protein TcpC (VF0413) combats the host's innate immune defense by abrogating the function of MyD88 in macrophages, inhibiting its signaling pathways (85) (Fig. 6). Thus, our observations suggest an interaction between some virulence factors with BMI and blood biochemical markers.

It has been shown that dysbiosis induces intestinal permeability, which associates

with systemic levels of bacterial products causing low-grade chronic inflammation, insulin resistance, and changes in plasma lipids (46). Furthermore, upregulation of cytokines and activation of innate immunity can result from excessive calorie intake, fat accumulation, lipotoxicity, and bacterial virulence factors (86). Pathogenicity is given by the combination of virulence factors and host physiology, where the pathobiome concept represents a community-wide approach that encompasses pathogenic and potentially pathogenic agents within their biotic environment (87, 88). In this study, we describe associations between microbiome taxa, functional capacity, and virulence factors with obesity, metabolic parameters, and dietary intake; supporting the notion that virulence factors and pathofunction, which are specific features of host bacterial communities, have the potential to play a role in non-communicable diseases development (87).

**Conclusion.** The onset of obesity is a complex process that involves the gut microbiome composition, diet, and age, among others. Changes in taxa's membership, relative abundance, and physiology, lead to different metabolite profiles significantly affecting host physiology (67). Childhood obesity is a known risk factor for obesity in adults (89). In addition, infancy is a critical period in the development of commensal gut bacteria, since its resilience varies throughout the child's maturation (3, 12). Therefore, attention must be focused on identifying modifiable early life exposures and preventive strategies associated with obesity risk in childhood.

In the urban Mexican children population, dietary patterns seem to be driving major differences in the microbiome's taxonomical composition and functional capacity between individuals with normal weight and overweight or obesity. We found taxa and functional pathways associated with anthropometric data, metabolic parameters, and dietary intake. Gut microbiome of children with OWOB in this study showed higher taxonomy diversity; whereas its virulence factors diversity was reduced compared to the NW. In addition, we found that lipooligosaccharides were the predominant factors in the gut microbiome of children with OWOB, corresponding to the loss of diversity and evenness in this group. Virulence factors were mainly associated with the population's metabolic parameters. Cross talk between host and microbiota, combined with environmental factors that impact its relationship, such as diet, opens a whole array of possibilities for preventing metabolic impairments in childhood, starting with overweight and obesity.

Because of the small sample size, only 2 communities (NW and OWOB) were compared in this study. Although this comparison reflects organismal and functional differences, including an overweight group in this study could provide greater insight into the dynamics and transition of these abnormalities. Other restrictions are connected to the frequency with which food is consumed. However, this nondifferential error is unlikely to affect the validity of our findings. More research is needed to stratify people using tailored predictions and find microbial biomarkers linked to different dietary responses.

## MATERIALS AND METHODS

**Sample and data collection.** This cross-sectional study is part of a previously reported study (31) approved by the Research Committee of the National Institute of Public Health (INSP, No.1129). A randomized selection of 48 samples was performed out of a biological bank containing 1042 samples (fecal samples). The reads from 3 samples failed to annotate correctly against the taxonomy and functional reference databases, excluding them and leaving a sample size of $n = 45$. Informed consent and assent, before the inclusion and use of biological samples and data, were obtained from all subjects and their parents or legal guardians involved in the study. After obtaining permission and informed consent from all participants and their parents, a FFQ and anthropometric measurements were conducted. Children with infectious or gastrointestinal disorders, as well as those who had taken antibiotics over the previous 2 months, were excluded from the study. Parameters, such as mean, standard deviation, confidence interval, median, minimum and maximum quartiles, and normality, were calculated. Mean comparison tests (wilcoxon rank-sum test and $t$ test) were used to explore significant differences between BMI groups of variables. Statistical significance was considered with $P$ value $< 0.05$.

**Adiposity.** The who2007 function in the RStudio program was used to calculate z-scores of the BMI adjusted for age and sex in children aged 5 to 19 (32). The reference tables of weight, height, and BMI adjusted by age of the R package were used.

**Dietary Patterns.** Each participant's baseline dietary consumption was assessed using a validated FFQ. This survey consisted in frequency consumption of 11 food categories and was completed in the presence of

their parents or tutor. Based on nutritional properties, the 107 food items comprised in the FFQ were grouped into 27 food groups (Table S1). The daily intake average was estimated based on the reported frequency of each item among the groups (10 options ranging from never to ≥6 times per day). The acquired factor was multiplied by the equivalent grams that make up a portion based on the Mexican National Health and Nutrition Surveys (ENSANUT) (90) to get the number of grams that each child consumed daily of each meal. The percentage contribution of each of the 27 food groups to the total daily consumption was estimated using grams or milliliters ingested by each participant, which were then normalized using z-scores (Table S2). After confirming that the quantitative normalized dietary data was correlated using the corrplot R package (v 0.84), we performed a principal component analysis (PCA), which provided 2 components with a threshold of 2.8253 eigenvalues, explaining 23.7% of the total variance of the individual's diet (Fig. S1). An orthogonal rotation (varimax) was used to redistribute the explained variance and acquire the most extreme weight factor, as well as to separate the 2 components to perform a better interpretation of them (Table S3). We considered the loading factors (eigenvectors) for each group of food calculated, for each of the 2 components, lower than -0.4 and greater than 0.4, to contribute significantly to the dietary pattern. The 2 dietary patterns were classified as "Pattern 1," which included protein and complex carbs, and "Pattern 2," which included saturated fat and simple carbohydrates. Finally, an individual score was generated for each factor, assigning each participant a value for each dietary pattern. The highest value between both dietary pattern's score was interpreted as the most likely dietary type consumed by each individual (Table S4).

**Genomic DNA extraction.** The QIAamp DNA Stool minikit (Qiagen) was used to extract DNA from 200 mg of fecal matter. QIAamp kit buffers, proteinase K, and high temperature incubation were used to lyse bacteria. Wash and purification columns were used to recover released genomic DNA. The isolated DNA was kept at –20°C until it was needed.

**Metagenomic sample processing.** Sample processing was carried out as previously stated (42). Briefly, pair-end libraries were created from 1 ng DNA, strands were enzymatically fragmented with ATM to 300 bp to 600bp, and unique barcodes were added to each sample following Nextera XT protocol (REF: 15032350, Illumina). The enrichment PCR was performed with an initial denaturation for 3 min at 72°C and 30 s at 95°C, followed by 12 cycles of denaturation for 10 s at 95°C, annealing for 30 s at 55°C, and elongation for 30 s at 72°C, and a final elongation step for 5 min at 72°C. All libraries were purified with Ampure XP magnetic beads (A6388, Beckman Coulter), following the manufacturer's recommendations. The quantity and quality of each library was assessed with the Qubit dsDNA HS assay kit (Q32851, Thermo Fisher Scientific), and a DNA Bioanalyzer (5067-4626, Agilent Technologies), respectively. Equimolar ratios of each sample were pooled and sequenced at the USec from National Institute of Genomic Medicine (INMEGEN) in 2 separate runs on the Illumina NextSeq technology, yielding 1,624,041,934 total pair-end reads 150 bases long.

**Metagenomic analysis: sequence quality and filtering.** FastQC (version 0.11.8) (91) was used to assess the quality of raw reads, followed by pre-processing with Cutadapt (version 1.18) (92), to removed barcodes, bad quality bases (<20 Phred score), and short length fragments (<20 paired bases). The 1,618,036,578 high quality reads were filtered for host contamination by mapping them individually to the *Homo sapiens* genome assembly (Ensembl release 95, GRCh38.dna.alt) using Bowtie2 (version 2.3.4.3) (93). Human and microbiome reads were identified and separated using Sam tools view (version 1.9) (94), with -f 4 for unmapped reads and -F 4 for mapped reads (custom perl script).

**Metagenomic annotation.** High quality 1,617,251,896 microbial paired-end reads were processed using the HMP Unified Metabolic Analysis Network (HUMAnN version 3.0.0, http://huttenhower.sph.harvard.edu/humann) pipeline, using for alignment the ChocoPhlAn and UniRef90 complete databases (95, 96). Richness (presence/absence) and abundance of microbial pathways for each of the 45 metagenomic samples was performed to obtain functional profiling. In addition, metagenomic phylogenetic analysis (MetaPhlAn, version 3.0) was used to perform taxonomic and organism-specific functional profiling, using unique clade-specific marker genes from reference genomes (95). Gene family abundance reported in reads per kilobase (RPK) units to normalize for gene length, computed as the sum of the scores for all alignments for a gene family, was normalized using copies per million (CPM) and relative abundance.

**Taxonomy and functional profile analysis.** Composition analysis of microbial communities from the 45 sample metagenomic shotgun sequencing data were obtained using MetaPhlAn 3.0. Alpha and beta diversity were calculated using Vegan Community Ecology R Package (version 2.5 to 6). The normalcy test was assessed using the Shapiro-Wilk test, and further difference between NW and OWOB groups was obtained using Wilcoxon rank-sum test or T-test, for non-parametric and parametric data, respectively. NMDS analyses were performed to evaluate distances between groups. Stress values obtained were <0.2 for Bray Curtis, Jaccard, and weighted Unifrac beta diversities. We evaluated beta diversity statistical differences and associations between microbiome composition and potential confounders applying analysis of variance using distance matrices with Adonis (PERMANOVA) from the Vegan R package. To obtain a list of species that accumulated explained more than 70% of the variation between groups, we used the Vegan R package function simper with 999 permutations. Wilcoxon rank-sum Test was used to evaluate differences between groups of each taxon obtained by simper. Continuous variables were fitted as a vector on the NMDS analysis using the envfit function of the Vegan package (97). We used LDA effect size (LEfSe) to characterize differentially abundant taxonomic and functional features (*P* value < 0.05) between NW and OWOB groups (34). To understand how microbial functional capacities further relate to our population's metabolic and dietary parameters, we tested MetaCyc pathways and gene families as summarized by level-4 EC categories obtained through HUMAnN3 (36, 37). Pathway coverage file details the richness (presence/absence) of the pathways in a community. Reactions detected in the community are assigned a confidence score, based on whether their abundance is greater or below the median reaction abundance. Pathway abundance file details the abundances of the pathway's component reaction, which is computed as the sum overabundances of genes catalyzing the reaction. Identification of a parsimonious set of pathways that explain observed reactions in the community was

performed using MetaCyc pathways definitions and MinPath. Gene families file details the abundance of each group of evolutionarily-related protein-coding sequences with similar functions in the community. The abundance of a gene family is stratified to show the contributions from species as a result of MetaPhlAn and ChocoPhlAn databases and translation. Gene family abundance is reported as reads per kilobase units (RPK) units to normalize for gene length, reflecting relative gene copy number in the community.

**Genetic elements of pathogenicity.** ABRicate v1.0.1 (Seeman T) was used to screen for virulence factors with the virulence factor database (VFDB supplemented with metagenomic sequences; database date October, 2020) (98). Sequences were aligned to this virulence factor repository setting a coverage criterion (–mincov) of ≥75% and identity criteria (–minid) of ≥50%. Richness and alfa diversity between NW and OWOB groups was analyzed using Phyloseq v1.30.0 (99). Read counts data were analyzed for differential abundance distribution using edgeR Bioconductor package (version 3.28.0), performing the generalized linear model methods (glmFit and glmLRT) (100, 101). Normalization for virulence factors composition effect was performed in order to compare relative changes in abundance levels between annotations using the trimmed mean of M-values (TMM) between each group of samples (102). Negative binomial generalized linear models were fitted, and tagwise dispersion estimates were obtained using the Cox-Reid profile-adjusted likelihood method to determine differential expression. Differentially abundant virulent factors were obtained after filtering by logarithmic fold change 1 and $-1$, $P$ value $\leq 0.05$, and false discovery rate (FDR adjusted-P) $\leq 0.05$, yielding 84 factors in total.

**Microbiome features associations with population metadata.** To identify individual microbial features (taxa, functions, and virulence factors) associated with anthropometric, serologic, and diet covariates, we assessed each feature's abundance using multivariate linear modeling implemented by MaAsLin2 (103), correcting for age and sex, or physical activity, depending on the analysis. These analyses used transformed relative abundance data and implemented sparse (zero-inflated) models for compositional microbial feature data correction. We required a minimum relative abundance of 0.01% for taxa. MaAsLin2 default parameters were used under the compound Poisson linear model (CPLM) analysis method, accounting for non-count (continuous distributed relative abundance) data with exact zeros regularly arising; we accounted for multiplicity of tests using the BH correction (104). Virulence factors clustered by VFID were further analyzed using the quasi-likelihoodF-test (Qlf) in edgeR (v. 3.28.1) in order to obtain the differential presence of virulence factors between BMI groups (logarithmic fold change over 1 and under $-1$, $P$ value and FDR $< 0.05$) (100, 101). The association network was performed using the coefficients resulting from the CPLM analysis. Each virulence factor category was associated with the anthropometric, metabolic, and dietary characteristics if the regression coefficient was statistically significant. We used a costume R script and the package igraph (version 1.3.5) to plot the network.

**Data availability.** The data reported in this paper are accessible in the NCBI Short Read Archive (SRA) under accession ID PRJNA721692. Original R scripts and data used for the analysis of this study are available in GitHub (https://github.com/sofiamurga/Microbiome-Taxa-FC-VF).

## SUPPLEMENTAL MATERIAL

Supplemental material is available online only.
**SUPPLEMENTAL FILE 1**, PDF file, 2.5 MB.
**SUPPLEMENTAL FILE 2**, XLSX file, 2.8 MB.

## ACKNOWLEDGMENTS

We thank all participants and Alfredo Mendoza Vargas from the Unidad de Secuenciación e Identificación de Polimorfismos, Instituto Nacional de Medicina Genómica (INMEGEN) for providing sequencing and support services, and the University of Wisconsin Center for High Throughput Computing (CHTC) in the Department of Computer Sciences for providing computational resources, support, and assistance. Finally, we would like to make an acknowledgment for the postdoctoral fellowship of F.C.-G. (CVU 443238) as part of the Estancias posdoctorales por México 2022 program from CONACYT.

S.M.M.-G., A.L.-M., T.P.-T., and A.I.B.G. conceptualized the study. S.M.M.-G., E.J.U.-P., Y.C.O.-O., A.L.-M., C.E.D.-B., F.C.-G., A.O.-L., and A.S.-F. performed the methodology. Formal analysis was conducted by S.M.M.-G., E.J.U.-P., and A.I.B.G. S.M.M.-G., A.L.-M., and A.I.B.G. investigated the study. A.I.B.G., A.L.-M., and M.C. took care of the resources. S.M.M.-G. and A.S.-F. curated the data. wrote the original draft Preparation, S.M.M.-G., A.L.-M., and A.I.B.G. wrote the original draft and prepared the manuscript. S.M.M.-G., E.J.U.-P., T.P.-T, Y.C.O.-O., M.C., A.O.-L., F.C.-G., C.E.D.-B., A.S.-F., A.I.B.G., and A.L.-M. wrote, reviewed, and edited the manuscript. The visualization was conducted by S.M.M.-G. and A.L.-M. A.I.B.G. and A.L.-M. supervised the research. A.L.-M. and A.I.B.G. administered the project. A.I.B.G. acquired the funds.

This research was funded by CONACYT SSA/IMSS/ISSSTE-CONACYT grant number 2015-262133, and CONACYT grant number CB2017-2018, A1-S-33221. S.M.M.-G. was

supported by "El Consejo Nacional de Ciencia y Tecnología – CONACYT" (The National Council for Science and Technology) with register number 895733.

The study was conducted according to the guidelines of the Declaration of Helsinki. The Biosafety and Research Commission of the National Institute of Public Health approved the project CB:1120-CI:1129 (INSP, Cuernavaca Morelos, México). Ethical approval on the project was given by the INSP Commission of Ethics (CI:1129-No.1294) after review of written consent.

We declare no conflicts of interests.

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
