## [Reviewer comments · Microbiology Spectrum]

Microbiology Spectrum

Virulence factors of gut microbiome are associated with BMI and metabolic blood parameters in children with obesity

Sofía Murga-Garrido, Ernesto Ulloa-Pérez, Cinthya Díaz-Benítez, Yaneth Orbe-Orihuela, Fernanda Cornejo-Granados, Adrian Ochoa-Leyva, Alejandro Sanchez-Flores, Miguel Cruz, Ana Castañeda-Márquez, Tanya Plett-Torres, Ana Burguete-García, and Alfredo Lagunas-Martínez

Corresponding Author(s): Alfredo Lagunas-Martínez, Instituto Nacional de Salud Publica Centro de Investigaciones sobre Enfermedades Infecciosas

Review Timeline:

Submission Date:	August 31, 2022
Editorial Decision:	November 10, 2022
Revision Received:	January 23, 2023
Accepted:	January 29, 2023

Editor: Jennifer Auchtung

Reviewer(s): Disclosure of reviewer identity is with reference to reviewer comments included in decision letter(s). The following individuals involved in review of your submission have agreed to reveal their identity: Jaime Garcia-Mena (Reviewer #1); Tian Liang (Reviewer #2)

Transaction Report:

DOI: <https://doi.org/10.1128/spectrum.03382-22>

November 10, 2022

Dr. Alfredo Lagunas-Martínez
Instituto Nacional de Salud Pública
Cuernavaca
Mexico

Re: Spectrum03382-22 (Virulence factors of gut microbiome are associated with BMI and metabolic blood parameters in children with obesity)

Dear Dr. Alfredo Lagunas-Martínez:

Thank you for submitting your manuscript to Microbiology Spectrum. Your manuscript was reviewed by two scientists with expertise in the field. Both reviewers indicated that revision was needed to improve readability. Here is a link to suggested English language services that could be used: <https://journals.asm.org/language-editing-services>.

While both reviewers had comments for how to improve your manuscript, some of reviewer 2's comments were quite broad. I have provided some clarifying suggestions to reviewer 2's comments 1 and 2 that should be addressed in a revised manuscript below. Please address all the reviewer comments in your revised manuscript.

Link Not Available

Sincerely,

Jennifer Auchtung

Journals Department
Reviewer comments:

Reviewer #1 (Comments for the Author):

The submission entitled "Virulence factors of gut microbiome are associated with BMI and metabolic blood parameters in children with obesity" describes an interesting research work, focusing on the characterization of the gut microbiota diversity, as well as the gut microbiome in a subsample of normal weight and overweight/obese children. In general, the text is well-written,

and the results are sound to the applied methodology and scope of the work. The listing of all identified taxa in the results section might be grouped by phylum since all the relevant names appear in the figures and/or legends. The discussion would have a clearer understanding for the readers if the authors could draw a figure with the main results of relevant taxa, metabolic pathways, and dietary and anthropometric data.

Reviewer #2 (Comments for the Author):

1 The manuscript is too long and poorly readable, further reductions are recommended.

2 Too many figures in the result and the quality is uneven, e.g. FIG3. It is suggested that the figures should be put together, which will make your paper more beautiful and flow.

3 FIG5 B has very obvious splice marks, need you to explain why this is.

4 As a whole, the manuscript is just performing a correlation analysis, perhaps you can try more other analysis tools, such as introducing network analysis to explore the interaction between clinical indicators and microbial abundance

5 In FIG1, it is not very meaningful to show the species composition of a single sample, and it is recommended to compare the species composition by different groups.

6 The section on dietary information survey is relatively well worked.

7 Why use only a few clinical indicators? AST, ALT, liver ultrasound and other information are more meaningful for NAFLD studies.

8 The manuscript is overloaded with references and many of them are unnecessary as well as inaccurately cited for a relatively long time. It is recommended that the references be removed and updated.

9 Finally, the manuscript needs further polishing and revision.

Editor's clarifying comments for Reviewer 2 comments 1 and 2:

R2 Comment 1: The manuscript is too long and poorly readable, further reductions are recommended.

Editor comment: The suggestions offered by reviewer 1 - "The listing of all identified taxa in the results section might be grouped by phylum since all the relevant names appear in the figures and/or legends. The discussion would have a clearer understanding for the readers if the authors could draw a figure with the main results of relevant taxa, metabolic pathways, and dietary and anthropometric data." - could provide guidance for how to address these concerns. Specifically, when taxa are clearly shown in figures they do not need to be listed in text. Examples include:

Line 179-184: Rather than listing all "marginally differential" taxa, it would be preferred to indicate these taxa are shown in Figure S2.

Line 227-230, 237-240, 243-248

R2 Comment 2 Too many figures in the result and the quality is uneven, e.g. FIG3. It is suggested that the figures should be put together, which will make your paper more beautiful and flow.

Editor comment: Several figures that focus on a similar set of comparisons could be coupled together. For example, figure 1-3 could be part of a multi-panel figure as could figures 4 and 5. I agree with the reviewer that figure 3 is currently of poor quality and not interpretable. NMDS plots should show the distinct data points from each sample that contribute to the clustering and indicate the stress of the plot, which allows the reader to interpret how these two dimensions shown represent the complexity of the multi-dimensional data. Figure 6, which shows Firmicutes to Bacteroidetes ratio, should be removed, as consensus in this measure as a relevant marker of obesity has declined (e.g., as reviewed in doi: 10.3390/nu12051474). An additional pair of figures that could be combined are Figures 10 and 11.

Staff Comments:

Preparing Revision Guidelines

Please return the manuscript within 60 days; if you cannot complete the modification within this time period, please contact me. If you do not wish to modify the manuscript and prefer to submit it to another journal, please notify me of your decision immediately so that the manuscript may be formally withdrawn from consideration by Microbiology Spectrum.

Response to Reviewers' Comments:

Reference manuscript: Spectrum03382-22

Reviewer #1 (Comments for the Author):

The submission entitled "Virulence factors of gut microbiome are associated with BMI and metabolic blood parameters in children with obesity" describes an interesting research work, focusing on the characterization of the gut microbiota diversity, as well as the gut microbiome in a subsample of normal weight and overweight/obese children. In general, the text is well-written, and the results are sound to the applied methodology and scope of the work.

- 1. The listing of all identified taxa in the results section might be grouped by phylum since all the relevant names appear in the figures and/or legends.**

We thank the reviewer for his/her suggestion. We have grouped the taxa listed to improve text readability in the following lines:

Lines 171-173: Other taxa marginally differential between both groups are shown in Supplemental Figure S2 (LDA scores (log10) <-2 and >2, 0.1 >p-value > 0.05).

Lines 209-213: The abundance of species included in the genera *Victivallis*, *Prevotella*, *Mitsuokella*, *Clostridium*, and the family Ruminococcaceae were overall associated with decreased values of the population's parameters. Whereas, members of *Parvimonas*, *Klebsiella*, *Gemella*, *Enterococcus*, and *Citrobacter* showed positive association with waist:hip ratio, blood glucose, triglycerides and total cholesterol (Figure 4).

Lines 221-229: We found species of the genera *Prevotella*, *Megasphaera*, *Klebsiella*, *Gemella*, *Citrobacter*, *Bacteroides*, and *Alloprevotella* positively associated to total energy intake, sugar, lipid, saturated-fatty acids, and trans-fatty acids percentage intake (Figure 4). Interestingly, the abundance of most of these taxa were also associated with increments in the W:H ratio and the metabolic parameters, and lower protein, fiber, and unsaturated-fatty acids intake percentage. Conversely, we found members of *Victivallis*, Ruminococcaceae, *Mitsuokella*, and *Clostridium* associated with higher protein, fiber, carbohydrate, monounsaturated-fatty acids, and polyunsaturated-fatty acids intake (Figure 4).

Lines 230-234: Relevant taxa associated with protein and complex carbohydrates (pattern 1), or saturated fat and simple carbohydrates (pattern 2) intake included species of the taxonomical orders Lactobacillales, Clostridiales, Actinomycetales, Bacteroidales, Bifidobacteriales, Eubacteriales, Methanobacteriales, Desulfocivibrionales, Actinomycetales, and Pseudomonadales (Figure 4).

- 2. The discussion would have a clearer understanding for the readers if the authors could draw a figure with the main results of relevant taxa, metabolic pathways, and dietary and anthropometric data.**

We agree with this suggestion. We added a figure (figure 6) summarizing the relevance of the virulence factors associated with dietary, anthropometric and blood metabolic data, in host damage. We thank the reviewer; this addition has improved the manuscript.

Editor comment: The suggestions offered by reviewer 1 - "The listing of all identified taxa in the results section might be grouped by phylum since all the relevant names appear in the figures and/or legends. The discussion would have a clearer understanding for the

readers if the authors could draw a figure with the main results of relevant taxa, metabolic pathways, and dietary and anthropometric data." - could provide guidance for how to address these concerns. Specifically, when taxa are clearly shown in figures they do not need to be listed in text. Examples include: Line 179-184: Rather than listing all "marginally differential" taxa, it would be preferred to indicate these taxa are shown in Figure S2. Line 227-230, 237-240, 243-248

Reviewer #2 (Comments for the Author):

- 1. The manuscript is too long and poorly readable, further reductions are recommended.**

We thank the reviewer for his/her valuable suggestion. We have made reductions and editions through the whole manuscript to improve readability.

- 2. Too many figures in the result and the quality is uneven, e.g. FIG3. It is suggested that the figures should be put together, which will make your paper more beautiful and flow.**

We agree that rearranging the figures into panels gives fluence to the paper readability. We have made three multi-panel figures and 1 heatmap by putting together figures 1 to 3 (figure 1), figures 4 and 5 (figure 2), 8 and 9 (figure 4), and figures 10 and 11 (figure 5). Thank you for this appreciated suggestion.

- 3. FIG5 B has very obvious splice marks, need you to explain why this is.**

Figure 5B splice marks are a graphical reference to the scale (LDA score (log₁₀)) used by the linear discriminant analysis Effect Size (LEfSe).

- 4. As a whole, the manuscript is just performing a correlation analysis, perhaps you can try more other analysis tools, such as introducing network analysis to explore the interaction between clinical indicators and microbial abundance**

We thank the reviewer for his/her suggestion. We added figure 5C showing the association network between virulence factors clustered by VFID and anthropometric data, metabolic blood parameters and dietary intake obtained from FFQs. This addition has improved the manuscript.

- 5. In FIG1, it is not very meaningful to show the species composition of a single sample, and it is recommended to compare the species composition by different groups.**

Thank you for this suggestion. We modified figure 1 in order to compare taxonomical composition of both BMI groups.

6. The section on dietary information survey is relatively well worked.

We thank the reviewer for this comment, the dietary section has been modified to improve its understanding.

7. Why use only a few clinical indicators? AST, ALT, liver ultrasound and other information are more meaningful for NAFLD studies.

We used the clinical indicators of waist:hip ratio, body mass index, blood pressure, blood glucose, blood triglycerides, and blood total cholesterol because we aimed to collect an overview data of the metabolic health of the children population.

Although further investigation on the hepatic diseases associated to obesity in childhood in Mexican population would be interesting and necessary; this was not the aim of this work and therefore we, unfortunately, did not count with any liver biomarker measurements nor imaging technique (AST, ALT, liver ultrasound, NAFLD Fibrosis Score, platelet count, MRI, etc.). Moreover, nonalcoholic fatty liver disease (NAFLD) in children is associated with several extrahepatic manifestations were hyperlipidemia, insulin resistance, and obstructive sleep apnea are included (Sweeny & Lee, 2021). Overall, our population is considered to be in the healthy spectrum. The overweight and obese subpopulation, based on the biochemical parameters measured, appears to be in the early transition from this healthy stadium to a metabolically disorder. We did not find differences in blood triglycerides, total cholesterol, nor glucose between both BMI groups. Also, because the prevalence of NAFLD in pediatric population is estimated to be around 10% (Sweeny & Lee, 2021), we do not expect to have had enough power to assess NAFLD with the data collected in this study.

Katherine F. Sweeny, Christine K. Lee, Nonalcoholic Fatty Liver Disease in Children. 2021. *Gastroenterology & Hepatology*. 17:12

8. The manuscript is overloaded with references and many of them are unnecessary as well as inaccurately cited for a relatively long time. It is recommended that the references be removed and updated.

We have edited, removed and updated the references. Thank you for the comment, the manuscript has improved.

9. Finally, the manuscript needs further polishing and revision.

We have revised and edited the whole manuscript and it has improved, definitely. Thank you.

Editor's clarifying comments for Reviewer 2 comments 1 and 2:
R2 Comment 1: The manuscript is too long and poorly readable, further reductions are recommended.

R2 Comment 2 Too many figures in the result and the quality is uneven, e.g. FIG3. It is suggested that the figures should be put together, which will make your paper more

beautiful and flow.
Editor comment: Several figures that focus on a similar set of comparisons could be coupled together. For example, figure 1-3 could be part of a multi-panel figure as could figures 4 and 5. I agree with the reviewer that figure 3 is currently of poor quality and not interpretable. NMDS plots should show the distinct data points from each sample that contribute to the clustering and indicate the stress of the plot, which allows the reader to interpret how these two dimensions shown represent the complexity of the multi-dimensional data. Figure 6, which shows Firmicutes to Bacteroidetes ratio, should be removed, as consensus in this measure as a relevant marker of obesity has declined (e.g., as reviewed in doi: 10.3390/nu12051474). An additional pair of figures that could be combined are Figures 10 and 11.

January 29, 2023

Dr. Alfredo Lagunas-Martínez
Instituto Nacional de Salud Pública Centro de Investigaciones sobre Enfermedades Infecciosas
Cuernavaca
Mexico

Re: Spectrum03382-22R1 (Virulence factors of gut microbiome are associated with BMI and metabolic blood parameters in children with obesity)

Dear Dr. Alfredo Lagunas-Martínez:

Your manuscript has been accepted, and I am forwarding it to the ASM Journals Department for publication. You will be notified when your proofs are ready to be viewed.

Sincerely,

Jennifer Auchtung
Editor, Microbiology Spectrum
